# Microalgae as Sources of High-Quality Protein for Human Food and Protein Supplements

**DOI:** 10.3390/foods10123002

**Published:** 2021-12-04

**Authors:** Yanwen Wang, Sean M. Tibbetts, Patrick J. McGinn

**Affiliations:** 1Aquatic and Crop Resource Development Research Centre, National Research Council of Canada, 550 University Avenue, Charlottetown, PE C1A 4P3, Canada; 2Aquatic and Crop Resource Development Research Centre, National Research Council of Canada, 1411 Oxford Street, Halifax, NS B3H 3Z1, Canada; sean.tibbetts@nrc-cnrc.gc.ca (S.M.T.); patrick.mcginn@nrc-cnrc.gc.ca (P.J.M.)

**Keywords:** microalgae, protein content, protein quality, protein quality assessment, environmental factors

## Abstract

As a result of population growth, an emerging middle-class, and a more health-conscious society concerned with overconsumption of fats and carbohydrates, dietary protein intake is on the rise. To address this rapid change in the food market, and the subsequent high demand for protein products, agriculture, aquaculture, and the food industry have been working actively in recent years to increase protein product output from both production and processing aspects. Dietary proteins derived from animal sources are of the highest quality, containing well-balanced profiles of essential amino acids that generally exceed those of other food sources. However, as a result of studies highlighting low production efficiency (e.g., feed to food conversion) and significant environmental impacts, together with the negative health impacts associated with the dietary intake of some animal products, especially red meats, the consumption of animal proteins has been remaining steady or even declining over the past few decades. To fill this gap, researchers and product development specialists at all levels have been working closely to discover new sources of protein, such as plant-based ingredients. In this regard, microalgae have been recognized as strategic crops, which, due to their vast biological diversity, have distinctive phenotypic traits and interactions with the environment in the production of biomass and protein, offering possibilities of production of large quantities of microalgal protein through manipulating growing systems and conditions and bioengineering technologies. Despite this, microalgae remain underexploited crops and research into their nutritional values and health benefits is in its infancy. In fact, only a small handful of microalgal species are being produced at a commercial scale for use as human food or protein supplements. This review is intended to provide an overview on microalgal protein content, its impact by environmental factors, its protein quality, and its associated evaluation methods. We also attempt to present the current challenges and future research directions, with a hope to enhance the research, product development, and commercialization, and ultimately meet the rapidly increasing market demand for high-quality protein products.

## 1. Introduction

Over the last half century, factors including improved agricultural technology, improved food production, improved processing and supply chains, along with the increase in income per capita, have resulted in substantial reductions in hunger worldwide, even though the world population has almost doubled during the same period [1,2]. However, the global population is still increasing at a rapid pace and is estimated to reach 9.7 billion by 2050: an increase of more than 25% from 7.7 billion in 2019 (United Nations, 2019). To feed this growing population, an increase of 70% in food production is required [2]. Improved land clearing and the more efficient use of existing croplands have certainly contributed to the increased demand for crop production; however, the environmental impacts and trade-offs of these alternative means of agricultural expansion are under scrutiny [2,3]. Recent studies suggest that global food production is having huge environmental impacts on the planet and generates 30–34% of global greenhouse gas (GHG) emissions [4,5]. The methods and technologies conventionally used for intensifying agriculture have been very successful throughout the green revolution but are predicted to no longer be sustainable solutions soon. These strategies come with high-impact trade-offs on the environment, such as the disruption of natural habitats, threats to biodiversity, climate changing GHG emissions, deforestation, desertification for livestock production, and polluting nutrient run-off from chemical fertilizers, which are damaging aquatic and terrestrial ecosystems [3,6,7]. In addition to the growing population, many other factors have placed higher demands on finite global resources to provide more food and different types of food [8]. Significant changes globally, including rising incomes, growing cities and urbanization, aging populations, and the heightened awareness of the health impacts of what we consume, have been driving a shift in consumption patterns, of which the increasing dietary protein intake is of particular interest [9]. New technologies and products, along with changes in dietary patterns, such as a growing vegan population, can potentially help to reduce the environmental impact of food production [10,11,12]; this change has stimulated the evolution of agricultural policy and investment in research and product development from various governmental levels to the private sectors.

Consumers are increasingly aware of the impact of specific nutrients on health and disease and this trend has, in turn, been driving food market growth, in particular the sector of functional foods, dietary supplements, and nutraceuticals. It is emerging that consumption of fat and carbohydrates is associated with the onset and progression of metabolic diseases, while protein intake, generally, is not [13,14,15]. Consistently, the global market for protein ingredients was USD 38.5 billion in 2020 and is anticipated to experience a compound annual growth rate (CAGR) of 10.5% from 2021 to 2028 (https://www.grandviewresearch.com/industry-analysis/protein-ingredients-market (accessed on 28 November 2021)). Animal-derived proteins have long provided significant human food security by supplying high-quality essential amino acids, dietary calories, and critical micronutrients, such as vitamins and minerals, which are often lacking in many terrestrial plant-based foods [16,17,18]. The impacts that particular foods (and their production systems) have on regional and global food security, and overall ecological sustainability, are measured by numerous factors [19]. Many human food resources are also in demand for formulated livestock feeds, and most terrestrial farmed animals convert raw feedstuffs into food products with relatively low efficiency [16]. As a consequence, the required higher production volumes of terrestrial foods must strike a balance with its ecological footprint and nutrient conversion efficacy. Thus, protein-rich products derived from terrestrial livestock production are now under particularly strong scrutiny [20]. In this regard, high-quality plant-based proteins, derived from novel sources, such as pulses (e.g., peas, lentils, etc.), have become increasingly popular and have been recently taken as one of the key strategies to meet the fast-growing food protein market. Nevertheless, due to limited resources of arable land and fresh water, a strong demand for alternative plant proteins has arisen, resulting in the increased development and use of algal proteins.

In recent years, microalgae have become important crops for global food and beverage industries, aqua-farming, and animal and human nutrition. The reasons include the following: (1) high content of protein, essential amino acids, and other healthy nutrients such as vitamins, antioxidants, omega-3 PUFAs, and minerals; (2) long-term sustainability, because microalgae have the lowest carbon, water, and arable land footprints of any crops; (3) environmental pollution remediation (e.g., ecological services); (4) high productivity compared with terrestrial crops and animal foods [1,21,22,23]. Algal proteins have already been used as food items, animal and aquaculture feeds, dietary supplements, pharmaceuticals, and cosmetics. The global algal protein market is expected to grow at the same rate as the total protein market, to reach USD 0.84 billion by 2023, from USD 0.6 billion in 2018 [9]. However, the development and use of microalgae for human foods are still at very early stages [1]. Algal cultivation techniques are first developed at laboratory conditions, with a controlled environment, and then moved to outdoor conditions for large-scale production of biomass. Cultivation of microalgae at large scales for commercial production is complicated. In addition to skilled and experienced personnel, methods and protocols must be developed, targeting specific species, culture systems, production plant locations, and the assessment of product quantity and quality [24,25]. This review focuses on the use of microalgae as human food, microalgal protein quantity, the impact of analytical methods on microalgal protein content, influence of environmental factors on microalgae protein yield, protein quality, and methods for protein quality assessment.

## 2. Historical Use of Microalgae as Human Food

Microalgae consumption is not foreign to humans, as their first use dates back over 2000 years, when Chinese people used *Nostoc* to survive during famine; additionally, the use of microalgae by indigenous peoples has occurred for centuries [26]. Despite this long history, however, only a handful of wild microalgae species, such as *Arthrospira platensis* (*Spirulina*), *Chlorella vulgaris* (*Chlorella*), and *Aphanizomenon*, have been domesticated and grown for human consumption and/or use [1,10,27]. Indeed, technologies for intensive outdoor cultivation of microalgae has only been developed since the 1950s [28]. The earliest efforts to mass cultivate microalgae with what can be considered modern technological approaches were carried out in Japan during the period of economic recovery following World War II [29]. With some strains accumulating up to ~65–70% (*w*/*w*) protein, microalgae were considered a promising source at a time when the supply of plant and animal protein was limited and therefore expensive [7,30,31,32,33]. In photosynthetic microalgae, proteins accumulate to such high levels during periods of rapid cell division and growth, when the synthesis of large protein complexes that are required for light harvesting and carbon fixation functions are at or near maximum capacity [34]. Research into the mass cultivation of microalgae began in the late 1940s and early 1950s in the United States and Europe, in particular, in Germany [35,36]. Interestingly, most of the theoretical constraints which limit the growth and yields of microalgae in outdoor tank or pond cultures were understood by this time. Similarly, it was during this period that all of the practical barriers to mass cultivation of algae—related to, for instance, the supply of CO_2_ and essential nutrients and the requirement for turbulent mixing—were first encountered and solved [35]. Aside from a few incremental advances in some of the finer technical details, the basic design and operation of open-pond cultures of protein-rich microalgae has changed very little since the 1970s [36,37]. However, despite continued research on optimizing microalgae growth and biomass yields in open-pond cultures, and as a result of a variety of technical and economic reasons, to date, microalgae have not been used widely as a source of protein, but rather have mainly been propagated as a source of whole biomass [24].

Several microalgal species are currently exploited for a variety of biological and industrial applications, including human foods, functional ingredients, cosmeceuticals, pharmaceuticals, animal and aquaculture feeds, fatty acids, alginates, carotenoids, wastewater treatment, and biofuels [38,39,40,41,42,43,44,45]. The cyanobacteria *Arthrospira* sp. and *Chlorella vulgaris* are sold not only as protein-rich food ingredients and supplements, but also as functional foods due to their high vitamin, mineral, and carotenoid contents, and they are generally regarded as safe (GRAS) [39]. While annual global microalgae production is presently rather modest (5.0 × 10^4^ tonnes dry matter) compared with macroalgae (seaweeds) (7.5 × 10^6^ tonnes dry matter), microalgae biomass and bioactives extracted from it are of great nutritional and economic importance (values at USD 1.25 × 10^9^ annually) [38]. Importantly, microalgae represent a large, polyphyletic group, numbering tens of thousands of different species, the majority of which have not been studied, let alone commercialized. Therefore, a huge potential exists in exploring and developing microalgae as sources of high-quality, sustainable protein for human food and dietary supplements.

## 3. Protein Quantity and Difference among Microalgae Species

Microalgae are a very diverse group of microorganisms and are estimated to comprise approximately 200,000 species [46]. The most familiar and arguably more ecologically important microalgal classes/divisions are Cyanophyceae (blue-green algae), Chlorophyceae (green algae), Bacillariophyceae (including the diatoms), and Chrysophyceae (including the golden algae) [39]. Protein contents can be vastly different among microalgal species and strains (Table 1) and is substantially affected by the environments in which they are grown. Under cultivation, many species may contain high levels of protein, typically 40–60% of dry matter, while some species have relatively low levels of protein, especially those selected for oil and biodiesel production [34], similar to the values of many species that were reviewed by others [47]. It was reported in a review that the crude protein content in microalgae biomass ranges from 6 to 63%, where most species have over 40% crude protein content, on the basis of dry mass [39]. The analysis of protein content in 16 microalgae revealed that protein content (% of dry cell matter) ranged from 12% (*Chaetoceros gracilis*, a diatom) to 35% (*Nannochloropsis oculata*, a eustigmatophyte) [48]. In 2007, Becker provided an overview on the major constituents of 13 microalgae [34]. There was a large range of protein content, 6–71% of dry matter, with over 50% in most of the species. An exception existed in *Spirogyra* sp., which had a lower, but larger, range of protein content (6–20% of dry matter), in line with the values of 12–24% reported by Tipnee et al., (2015) [49] and 18% by Saragih et al., (2019) [50]. A similar range (10–71%) of protein content was reported in 2013 by Becker in 33 microalgal species [51]. In 2020, Acquah et al. reported a range of protein content of 6–58% in 17 microalgal species [47]. Another review presented the protein content of 22 microalgal species, studied in different laboratories. Except *Spirogyra* sp. (6–20%) and *Scenedesmus dimorphus* (8–18%), a range of 28–71% of protein in dry biomass was reported and most species had over 50% protein [52]. It is evident that most microalgal species contain a high content of protein and some of them have been developed and used for food proteins or dietary supplements, such as the chlorophyceae *Chlorella* sp. and *Scenedesmus obliquus* and the cyanobacteria *Arthrospira* sp. [34]. It appears that *Chlorella vulgaris* and *Arthrospira* sp. are the most commonly exploited industrial species due to their high protein contents (51–58% of dry matter) and favorable essential amino acid profiles [34,53].

It is understood that the differences in protein content between microalgal species are primarily attributed to their different genetic traits. However, the reason for different protein contents of a given species is multifaceted, including the method employed for protein analysis, the environmental conditions under which they were cultivated, and the growth phase at which the microalgal biomass was harvested.

## 4. Influence of Analytical Methods on the Protein Content of Microalgal Biomass

Several methods have been used to measure protein content, including nitrogen analysis, colorimetric assays, and summation of anhydroamino acids [48]. These methods generate different values of protein content for the same samples [48,70,71]. In 2010, Lopez et al. measured the protein content in the dry biomass of five microalgal species by analyzing total nitrogen, using the Kjeldahl method and elemental analysis [72]. It was found that the nitrogen-to-protein (N-to-P) conversion factors for biomass obtained from the exponential phase were 5.95 for nitrogen measured by Kjeldahl and 4.44 for nitrogen measured by elemental analysis, and the protein content in the dry biomass ranged from 30% to 50%. An N-to-P conversion factor range of 3.00–6.35 has been reported for various microalgal species that were analyzed using different methods and in different research laboratories [47]. Apparently, elemental analysis of total nitrogen generated a higher value of total nitrogen and thus a lower N-to-P conversion factor compared with the Kjeldahl method. A different N-to-P conversion factor of 4.22 was established by others, who also used the method of elemental analysis to measure the total nitrogen [73]. In this study, nitrogen content was analyzed in 12 microalgal species/strains cultivated under mixotrophic and autotrophic conditions, and the protein content ranged from 73.9–76.5% [73]. Another study reported the protein content of microalgae ranging from 7 to 40% and changing dramatically over the course of their growth cycles [74]. In this study, 100 samples of 3 species were analyzed for protein content, using elemental analysis, and an N-to-P conversion factor of 4.78, established by others [73], was used for the calculation of protein content. The elemental analysis has generated a narrow range (4.22–4.78) of N-to-P conversion factors. An average of 4.78 was recommended for estimating protein content when elemental analysis is employed [73], and was later confirmed to be appropriate when the high non-protein nitrogen (NPN) was taken into account [75]. However, it was emphasized that differences exist between species and strains and even between samples collected from a given species or strain under different growing conditions or harvested at different growth phases [75]. The difference in N-to-P conversion factors is considered to be a result of different concentrations of NPN-containing molecules, particularly nucleic acids. It was reported that the median composition of nutrient-sufficient, exponentially growing microalgae was 5.7% RNA and 1% DNA, and varied between samples cultured under different conditions and collected at different growth phases [76]. Thus, validation and optimization of the N-to-P conversion factor are important for accurately measuring the protein content of microalgal biomass. This can be implemented by referring to the results by other methods that are able to measure the “true” protein concentrations, such as the summation of anhydroamino acids [75,77].

The Lowry method is a colorimetric assay, based on both the Biuret reaction and the Folin–Ciocalteau reaction, results in a strong blue color, which depends partly on the tyrosine and tryptophan content [78]. Although standards are used for calibration, the standard protein may not match that of the protein of interest and many substances interfere with the reaction; in addition, Folin reagents are reactive for a short period of time after addition [79]. Although this method can be used for quantifying proteins [80], the accuracy largely relies on the accessibility to intracellular proteins, or in other words, on the efficacy of cell wall disruption and protein extraction from the raw biomass. As such, the accuracy of this method is highly dependent on the cell wall structure, its rigidity, and its protein extraction efficiency, which can be quite different between microalgal species. Therefore, this method may be used for the measurement of protein content in extracted or purified protein products but may not be suitable for measuring protein content in the raw biomass of microalgae. On the other hand, the colorimetric methods may be used for high throughput screening of multiple algal species and strains, and when an accurate analysis is required, a more reliable method must be employed.

Accurately determining protein content in the raw material of microalgae is critical to the valorization of algal biomass for food and other forms of application, as microalgae are currently used predominantly in the form of raw biomass, while increased use of the purer forms of microalgal protein products may be achieved in the near future. In this regard, the method of elemental analysis is well-accepted and widely used for its multiple advantages, including, but not limited to, its simplicity, rapidity, and inexpensiveness, while the accuracy depends on the use of an appropriate N-to-P conversion factor [81]. Therefore, there exists a demand to establish an N-to-P conversation factor in reference to the value generated with a more reliable method, such as the summation of anhydroamino acids [48]. As amino acids are the building blocks or constituents of a protein, the summation of anhydroamino acids can be applied across different microalgal species, on the condition that proteins are completely released and fully hydrolyzed to single amino acids in the amino acid analysis. Due to differences in the ratio of protein nitrogen to NPN and differences in contents in different microalgae species and strains, dedicated N-to-P conversion factors may be required for the accurate measurement of protein content when elemental analysis is employed [82]. This idea may also apply to the samples collected from the same species or strain grown under different conditions or harvested at different growth phases. More research is required to determine dedicated N-to-P conversion factors in the biomass of different microalgae species or strains and also for samples of a given species or strain that are grown under different conditions and harvested at different growth phases. A quality control sample, which refers to a designated sample available in sufficient quantities such that replicate data can be obtained over a long period of time, should be used to validate the analysis [82]. In addition, to meet the requirement for comparing the results between different laboratories, analytical performance should be calibrated using a globally accepted, standard protein, produced by an internationally certified reference material laboratory or supplier.

## 5. Influence of Growing Conditions on Microalgae Protein Content

Many studies have been conducted to evaluate the impact of environmental factors on microalgal growth and biochemical profiles, including protein. A recent study showed significant effects of culture conditions on the growth rate, biomass production, nutrient composition, and the content of protein and other bioactive compounds in five microalgae species [83]. In addition to genetic traits, protein content in microalgal biomass is affected by a number of factors in the culture system, including light intensity and spectrum, temperature, CO_2_, pH, and nutrient media composition; moreover, the effect of each factor varies greatly among different species [83,84].

Light is the energy source of phototrophic microalgae for growth and synthesis of biomolecules. In some species, for example, *Botryococcus braunii*, the increase in light intensity was negatively related to protein content [85], while in other species, such as *Nannochloropsis* sp. [86] and *Ankistrodesmus falcatus* [87], a positive relationship was observed. In an experiment with *Chlorella vulgaris*, *Desmodesmus* sp., *Ettlia pseudoalveolaris*, and *Scenedesmus obliquus*, a clear pattern of increase in biomass yield or cell growth rate, with increasing light intensity from 50 µE to 500 µE m^−2^ s^−1^, was observed [88]. Greenhouse bioreactor trials, conducted throughout changing seasons, demonstrated that protein content remained relatively constant with rising light intensity and temperature. Protein content in *Isochrysis* sp. and *Rhodomonas* sp. decreased when they were cultivated at high temperatures in a range of 25–35 °C, while protein content increased in *Prymnesiophyte* sp., *Cryptomonas* sp., and Chaetoceros sp., with the largest change occurring in *Isochrysis* sp. [89], in line with the results reported by others [90]. A study with 8 marine microalgae species showed a U-shape relationship between protein content and temperature from 10 to 25 °C in *P. pseudonana*, *P. tricornutum*, and *P. lutheri*, and a bell shape in *C. gracilis*, while protein content decreased in *C. simplex* and *I. aff. Galbana* and remained slightly increased in *D. tertiolecta* and *C. calcitrans* with the increase in temperature [91]. Response of the protein content of microalgae to temperature and light intensity of culture systems varies from species to species, without a consistent trend; therefore, the determination of species-specific responses to light density and temperature is vital for the maximal rate of protein production.

Microalgae can grow in a wide range of non-ideal CO_2_ concentrations and their CO_2_ fixation rates are 5–8 times higher than terrestrial plants, of which, 70–80% goes to biomass production [92,93]. There exists an interaction between biomass productivity and CO_2_ concentration in the culture medium [94,95]. For example, *Scenedesmus bajacalifornicus* was investigated for CO_2_ fixation capability and biomass chemical composition, where cultivation was carried out at CO_2_ concentrations ranging from 5 to 25%, while the temperature and light intensity were kept constant. The CO_2_ concentration had a significant impact on growth and biochemical profile, with maximal biomass productivity and protein content achieved at 15% CO_2_ [96]. In a non-axenic polyculture of native microalgae of 16 species, CO_2_ concentration in culture media showed a significant effect on the biomass yield, and the maximal protein content was observed at 400 mg L^−1^ [97]. A review of 17 studies showed a substantial effect of CO_2_ supply on the biomass productivity and protein content, and an interaction between CO_2_ concentration and microalgal species [98]. For maximal protein production, a dedicated CO_2_ concentration in the culture medium must be established for a given species.

Nitrogen represents a critical macronutrient that regulates the metabolism and consequently, the growth and biochemical composition of microalgae [99]. In *Chaetoceros* sp., grown in batch cultures, nitrogen from 0.5 to 1.0 g/L led to an increase in biomass yield from 1 to 2 g/L and protein content from 38 to 46%, while no further increases were observed when the nitrogen concentration exceeded 1.0 g/L [100]. In contrast, it was observed in *Isochrysis galbana* that cell growth and protein content decreased with diminishing nitrogen concentration when cultured at nitrogen concentrations ranging from 0 to 0.29 mg/L [101]. Similarly, nitrogen limitation or starvation changes photosynthetic activity and reduces the protein content of microalgae in *Parietochloris incise* [102], *I. galbana* [101], and many other microalgal species [103]. Microalgae are generally able to utilize various forms of nitrogen, including nitrate, nitrite, ammonium, and organic nitrogen sources, for example, urea [104]. Each nitrogen source is first reduced to the ammonium form and assimilated into amino acids through a variety of pathways [103]. Although the source of nitrogen can affect the gross biochemical composition, the protein content of microalgae is more strongly controlled by the growth phase than by the particular source of nitrogen used for growth [105].

The pH of the nutrient media is also an important consideration for microalgae cultivation as it directly affects solubility of nutrients and minerals in the media, injected CO_2_, and photosynthetic activity of the algal cells [106,107]. In flask cultivation, the optimal pH for *Chlorella sorokiniana* was reported to be approximately 6.0 when only accounting for cell growth and not considering the CO_2_ fixation efficiency [106]. In the same study, a flat panel airlift photobioreactor was used for scale-up cultivation at five different pH levels. It was found that biomass productivity decreased while protein content in the biomass increased with the increase in pH [106]. It was demonstrated that microalgae *Euglena gracilis* did not survive at pH < 4 and >8, while the highest growth rate was detected at pH 7 and the photosynthesis was the most effective at pH 6 [107]. Each microalgal species has an optimal pH range for biomass production and biochemical composition, which is narrow, and species and strain specific, based on studies in 10 microalgal species [108,109,110,111]. Some key factors that can influence the pH of the cultivation media include its nutrient composition and their respective buffering capacities, CO_2_ concentration, cultivation temperature, and metabolic activity of the algal cells themselves. The methods for controlling pH include CO_2_ injection, buffer addition, and acid/base adjustment, with the first two approaches being used commonly in algae cultivation [26,108,112]. In cultures, buffer solutions are used to reduce pH fluctuations. However, the use of large volumes of buffers may be impossible in a large scale or industrial production settings because of cost. Instead, in aerate cultures, pH control is achieved by pumping atmospheric air or CO_2_-enriched air through the medium [113].

## 6. Protein Content of Microalgae Collected at Different Growth Phases

During different growth phases, microalgal cells undergo changes in cell structure, composition, and nutrient content [114]. It was reported that protein content in the extracellular polymeric substances fraction, extracted from the surface of *E. texensis* cells at the end of the stationary phase, was eight times as much as that at the end of the exponential phase [115]. An experiment on the effect of the growth phase on biochemical composition of two strains of *Tisochrysis lutea* revealed that protein content per cell of each strain was significantly higher at the exponential phase than at the stationary phase, regardless of whether nitrogen was replete or reduced [116]. In another study with *Isochrysis* sp. (clone *T. ISO*), *Pavlova lutheri*, and *Nannochloropsis oculata*, the protein accumulation was enhanced during the exponential phase but started to decline during the stationary phase [117]. Protein accumulation in the biomasses of *C. vulgaris* and *N. gaditana* was maximized during exponential growth but declined for *C. vulgaris* four days into the stationary growth phase [118]; this reaffirms the important effect of the growth phase on final biomass protein content. An optimal harvest time in the exponential phase should be determined by assessing both the accumulation rate and total yield to achieve the maximal protein production rate.

## 7. Protein Quality of Microalgae Biomass

Protein quality is vital when evaluating what nutritional value a protein product can provide. Several methods have been developed and used to assess the protein quality of a given product, such as amino acid composition, amino acid score (AAS, Equation (1)), essential amino acid index (EAAI), chemical score (CS), biological value (BV), protein digestibility (either in vitro or in vivo), net protein utilization (NPU), protein efficiency ratio (PER), protein digestibility corrected amino acid score (PDCAAS, Equation (2)), and digestible indispensable amino acid score (DIAAS) [47,119]. AAS and PDCAAS are two of the most commonly used parameters to date in protein quality assessment, with DIAAS being adopted slowly in recent years to replace PDCAAS in some countries [120]. The FAO/WHO expert panel concluded, in 1989, that AAS was a suitable measure of protein quality [121], defined as follows:AAS (amino acid score) = mg of amino acid per g test protein/mg of the same amino acid per g reference protein(1)

Subsequently, PDCAAS was adopted to correct AAS for the true fecal protein digestibility of the test protein as measured in a rat assay, calculated as follows:PDCAAS (protein digestibility corrected AAS) = (mg of limiting amino acid per g of test protein/mg of same amino acid per g of reference protein) × fecal true digestibility percentage(2)

This reference protein had earlier been recommended by FAO/WHO in 1985 and was based on the essential amino acid requirements of infants and young children [122], which have been periodically updated since that time [123,124]. AAS reflects the balance of essential amino acids and the abundance of the first limiting amino acid in a protein product, relative to the reference protein; therefore, a protein with a higher AAS value should yield a higher protein utilization rate or bio-efficiency in the body. A protein-rich ingredient, having an AAS equal to or higher than one, indicates that all of its constituent essential amino acids meet or exceed dietary requirements of the target age group. In this regard, animal-derived proteins are generally the highest rated, due to their well-balanced essential amino acid contents, relative to human and animal dietary requirements [19,119]. For example, casein has the AAS values of 1.03–1.32 based on several studies [19,125,126]. Microalgae *Chlorella vulgaris* and *Chlorella sorokiniana* had AAS values of 1.10 and 1.16, respectively, while *Acutodesmus obliquus* had an AAS value of 0.86 [81]. Kent et al. (2015) reported that the microalgae *Nannochloropsis* sp., *Scenedesmus* sp., *Dunaliella* sp., and *Chlorophyta* sp. had AAS values of 0.98–1.05, and two microalgae products derived from *Spirulina* and *Chlorella* had AAS values of 0.81 and 0.92, respectively [127]. The essential amino acids profile and AAS of some microalgae are summarized in Table 2. EAAI is similar to AAS and has also been used to evaluate the balance of essential amino acids in microalgae proteins [128,129]. Information on the AAS of microalgal proteins is limited and more research should be conducted in the future.

Although AAS reflects the amino acid composition of a protein, it does not tell the true biological value that a given protein can provide after it is ingested due to the differences in protein digestibility of various protein products. To adjust this, more comprehensive methods, including PDCAAS, have been developed to assess the overall quality of proteins in a product, which involves the use of AAS and in vivo protein digestibility [131]. Two types of in vivo protein digestibility have been used for over a century, depending on whether the endogenous protein contribution to the fecal protein output is corrected for or not. Apparent protein digestibility (ADP) does not correct for the amount of endogenous protein contributed to the total of undigested fecal proteins and thus underestimates the digestibility of proteins in a given product. Therefore, true protein digestibility (TPD) has been developed to correct for the contribution of endogenous proteins to the total fecal protein output. As significant fermentation and metabolism of proteins, peptides, and amino acids by bacteria occurs in the large intestine, ileal APD and TPD have also been established by collecting and analyzing ileal samples for the amounts of undigested proteins. However, sample collection in this method has some difficulties and thus not been widely practiced to date, even although it has been highly recommended. Although APD has been used to assess protein quality in some countries, TPD becomes mandatory in many countries where research technologies and instruments are available to carry out the required experiments and analysis. PDCAAS is a method for assessing the quality of a protein, based on both the amino acid requirements of humans and their ability to digest it and has been used commonly worldwide [120]. In Canada, PER is still the official method for protein quality assessment while PDCAAS is also accepted [132]. Most protein products have PDCAAS values of lower than one, especially the plant [125] and microalgal proteins [81,133].

Wang et al., (2020) evaluated the protein quality of *Chlorella vulgaris*, *Chlorella sorokiniana*, and *Acutodesmus obliquus* using a rat bioassay [81]. The raw biomass of these 3 species had PDCAAS values of 0.63, 0.64, and 0.29, respectively, which are lower than that of animal proteins, such as egg, milk, and meat proteins, but the value of the first 2 species were comparable to, or better than, wheat and pulse proteins [134,135]. The same group also investigated the effect of mechanical cell wall disruption using micro-fluidization and found the PDCAAS of the disrupted biomass of these 3 species increased to 0.77, 0.81, and 0.46. The PDCAAS values of *Chlorella vulgaris* and *Chlorella sorokiniana* are higher than that of cooked pulses [135]. The observed low PDCAAS of microalgal protein is primarily a result of low TPD, as *Chlorella vulgaris* and *Chlorella sorokiniana* had AAS values of over 1.0 [81]. Indeed, while ADP values for *Chlorella vulgaris*, *Chlorella sorokiniana*, and *Acutodesmus obliquus* were 60.6%, 54.8%, and 32.6%, TPD values were all higher, at 64.7%, 59.3%, and 37.9%, respectively. Furthermore, mechanical cell wall disruption significantly increased both ADP values (72.9%, 70.5%, and 62.4%, respectively) and TPD values (77.5%, 74.9%, and 67.2%, respectively). This study reported, for the first time, on the protein quality marker PDCAAS of microalgae biomass for human food using the standard rat assay, the impact of mechanical cell wall disruption, differences between microalgal species, and the elemental analysis of nitrogen in dietary and fecal samples for the calculation of ingested and undigested proteins. Recently, Tessier et al. [133] evaluated the protein quality of biomass produced from the blue-green algae *Spirulina* using a rat model. These researchers quantified ^15^N concentrations in caecal contents 6 h post-prandial and obtained a TPD of 86.0% and a PDCAAS of 0.84. These values are higher than those obtained for *Chlorella vulgaris*, *Chlorella sorokiniana*, and *Acutodesmus obliquus*, using the fecal protein digestibility method [81] and may be indicative of the different cell wall recalcitrance between the various species studied. A range of 51–90% of in vivo protein digestibility was observed in rats, mice, and humans for the biomass of several different microalgal species and strains (e.g., *Arthrospira*, *Chlorella*, *Coelastrum*, *Nannochloropsis*, *Scenedesmus*, *Uronema*) over the past decades by different research groups [47]. As different species were assessed for protein quality using different methods in the two aforementioned studies, it is not fully possible to elucidate whether the differences in TPD and PDCAAS between these two studies are attributed to the nature of microalgal species, function and products of gut bacteria, and/or a result of the different model and assay protocols employed. However, differences in cell wall recalcitrance are already well documented between *Spirulina* and *Chlorella*. The DIAAS values are corrected for the standardized ileal digestibility of individual amino acids, rather than simply protein as a whole. The difference between PDCAAS and DIAAS lies in the fact that fecal digestibility, used in PDCAAS, may be affected by microbial degradation, while true ileal digestibility used in DIAAS more accurately represents the amount of amino acids absorbed in the gastrointestinal tract [136]. There is no information available about the DIAAS values of microalgal biomass or more pure forms of microalgal protein products for human foods; however, high in vivo APD of essential amino acids (>92%) and DIAAS values (>1) were recently reported for *Pavlova* sp. microalgae when measured in Atlantic salmon [137].

As discussed previously, there is a rapidly growing interest in finding new protein alternatives to animal-derived sources. Of particular interest in recent years have been plant-based resources such as pulses and microalgae [10,138]. The protein quality of microalgae such as *Chlorella vulgaris* and *Chlorella sorokiniana*, as determined by PDCAAS values, appears to be higher than that of pulses such as lentils, beans, peas, and chickpeas [125]. Microalgae have been traditionally consumed as dried whole cells for their health benefits in addition to their abundant supply of nutrients [139]. However, digestibility can be low for some species, so cell wall disruption using a variety of techniques and the selection of species with inherently low recalcitrance is critical for those consumed as whole cells. Although more costly to produce, digestibility and ultimate protein quality of microalgae can be greatly enhanced if consumed in more refined forms, such as protein concentrates or protein isolates [140]. As the protein of some microalgae species has good essential amino acid profiles or amino acid scores, increasing protein digestibility is critical to the enhancement of microalgal protein quality and nutritional and commercial values. Extraction and purification are options when new processing technologies and equipment are developed by adapting to the industry needs and become affordable and economically attractive.

Another important application of microalgal proteins in human nutrition is to balance the essential amino acids of plant and pulse proteins as they have different first limiting amino acids. Methionine and cysteine or tryptophan are the limiting amino acids in most pulses, such as beans, peas, and lentils [135], while histidine or isoleucine are the first limiting amino acids in microalgae such as *Chlorella vulgaris*, *Chlorella sorokiniana*, and *Acutodesmus obliquus* [81]. When they are combined or consumed together, a more balanced amino acid profile may be achieved and thus improve the overall bio-efficiency of amino acids of the pooled protein product.

## 8. Current Challenges and Future Research Directions

Microalgae are a source of renewable nutrition and there is a fast-growing interest in algae-based functional foods and dietary supplements in the form of whole biomass, such as *Chlorella* and *Arthrospira* or purified protein products. The quality of food protein products is determined mainly by protein content, amino acid profile, and digestibility (Figure 1). Elemental analysis appears to be suitable for the estimation of protein content of microalgal biomass, while more research needs to be performed in terms of developing and standardizing N-to-P conversion factors. Different methods are used to measure protein quality, of which, AAS and TPD are critical, and determine the endpoint protein quality markers of PDCAAS and DIASS. Although microalgae have been well recognized as green and sustainable sources of high-quality protein—compared with other sources of protein, especially terrestrial plant proteins—and are increasingly used globally as dietary supplements or food protein ingredients, information on the protein yield and its interaction with growing conditions, protein quality assessment, and processing technologies for protein quality improvement falls far behind those of animal and other plant-based proteins. There is a strong need to increase investment and efforts in the research and product development of microalgae.

Microalgal proteins can be used to balance dietary proteins in countries where animal proteins are insufficient, or where the balance of dietary essential amino acids is a big challenge. Microalgal proteins are also valuable for human health promotion, particularly in those who are vegans or vegetarians [16,17,18]. Microalgal proteins have many other applications as well, such as cosmeceutical, pharmaceutical, and animal/aquaculture feed aspects, which are beyond the scope of this review and should be addressed separately. The wide use of microalgae protein for food remains a challenge, largely due to a lack of scalable and cost-effective cultivation and knowledge gaps regarding harvesting and downstream processing [10]. The important cost factors are the capital costs of algal ponds and associated operational costs including those of mixing processes, the photosynthetic efficiency of systems, the growth media, and CO_2_. An optimization study on these cost factors sheds a light as a future direction to improve economic output [141]. As low-trophic aquatic organisms, microalgae can synthesize and/or bioaccumulate other metabolites, contaminants, and impurities that have the potential to cause harm upon ingestion [142,143]. As such, significant research and product monitoring efforts are required to fully characterize the microalgae of interest and to improve production methods in order to ensure the nutritional value and safety of the final consumer products and this may be achieved by production of concentrated or purified forms (e.g., protein concentrates or isolates) for human food and supplements. Continued advancements in microalgae research and technology should result in increased biomass production and assist society in meeting our ever-growing demand for high-quality sustainable protein-rich foods.

## Figures and Tables

**Figure 1 foods-10-03002-f001:**
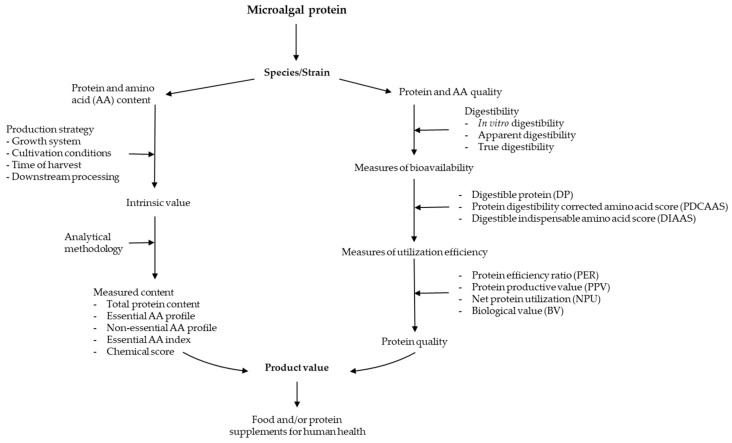
Factors affecting microalgal protein content and quality and common assessment criteria.

**Table 1 foods-10-03002-t001:** Protein content of different microalgal species.

Species	Protein Content (% Dry Matter)	Reference
*Acutodesmus dimorphus*	28	[54]
*Anabaena cylindrica*	43–56	[34]
*Aphanizomenon flos-aquae*	62	[34]
*Arthrospira fusiformis*	62	[55]
*Arthrospira maxima*	60–71	[31,34]
	65	[32]
*Arthrospira platensis* (Bangladesh)	60	[33]
*Arthrospira platensis* (France)	65	[33]
*Arthrospira platensis* (Malaysia)	61	[33]
*Arthrospira platensis* (Thailand)	55–70	[33]
*Arthrospira platensis*	63	[34]
	53–70	[7]
	45–62	[8]
	22–38	[9]
	61	[56]
	64	[55]
	17–32	[57]
	26–72	[58]
	22–51	[59]
	57–70	[60]
	45–62	[61]
	56	[54]
*Botryococcus braunii*	22	[62]
	39–40	[54]
*Chaetoceros calcitrans*	34	[48]
*Chaetoceros gracilis*	12	[48]
*Chlamydomonas rheinhardii*	48	[34]
*Chlorella 71105*	56	[63]
*Chlorella pyrenoidosa*	57	[34]
	60	[64]
*Chlorella pyrenoidosa and Chlorella vulgaris*	53	[54]
*Chlorella vulgaris*	51–58	[34]
	48	[56]
*Chroomonas salina*	29	[48]
*Diacronema vlkianum*	57	[1]
*Dunaliella hardawil*	10	[62]
*Dunaliella salina*	57	[34]
	29	[62]
*Dunaliella tertiolecta*	20	[48]
*Euglena gracilis*	39–61	[34]
*Haematococcus pluvialis*	48	[65]
*Isochrisis aff.galbana (T-iso)*	23	[48]
*Isochrysis galbana*	29	[48]
	27	[56]
*Nannochloris atomus*	30	[48]
*Nannochloropsis granulata*	18–34	[54]
*Nannochloropsis oculata*	35	[48]
*Nannochloropsis* spp.	29	[66]
*Neochloris oleoabundans*	30	[54]
*Nitzschia closterium*	26	[48]
*Nitzschia* sp.	17	[62]
*Nochloris oleoabundans*	30	[1]
*Pavlova lutheri*	29	[48]
*Pavlova salina*	26	[48]
*Phaeodactylum tricornutum*	30	[48]
	40	[54]
*Porphyridium aerugineum*	32	[54]
*Porphyridium cruentum*	28–39	[34]
	34	[67]

*Scenedesmus almeriensis*	47	[68]
*Scenedesmus obliquus*	50–56	[34]
*Skeletonema costatum*	25	[48]
*Spirogyra* sp.	6–20	[34]
*Spirogyra varians*	17	[49]
*Spongiococcum excentricum*	32	[64]
*Synechococcus* sp.	46–63	[34]
*Tetraselmis chuii*	31	[48]
	47	[54]

*Tetraselmis suecica*	31	[48]
*Thalassiosira pseudonana*	34	[48]
*Tisochrysis lutea*	37–42	[69]

**Table 2 foods-10-03002-t002:** Amino acid profile and score of different microalgal species.

Microalgal Species	His	ISO	Leu	Lys	SAA	AAA	Thr	Trp	Val	AAS ^#^	Ref.
Reference protein	16	30	61	48	23	41	25	6.6	40		[123]
*Acutodesmus obliquus*	**14** ^&^	36	85	41	33	92	59	20	60	0.86	[81]
*Acutodesmus obliquus* *	**12**	38	89	36	34	90	61	22	62	0.76	[81]
*Arthrospira maxima*	18	60	80	46	**18**	88	46	14	65	0.78	[34]
*Arthrospira platensis*	**22**	45	98	71	39	157	46	12	78	1.37	[130]
*Botryococcus braunii (A)*	**15**	34	71	47	39	72	37	22	44	0.94	[54]
*Chaetoceros calcitrans*	**19**	55	82	63	30	112	45	14	59	1.19	[48]
*Chaetoceros gracilis*	24	58	72	**51**	29	125	59	16	62	1.06	[48]
*Chlorella pyrenoidosa*	16	62	**34**	81	61	51	35		52	0.56	[128]
*Chlorella sorokiniana*	20	**35**	84	57	27	87	53	20	59	1.16	[81]
*Chlorella sorokiniana* *	19	**35**	83	58	28	86	50	22	59	1.16	[81]
*Chlorella vulgaris*	**18**	36	92	52	29	98	43	23	58	1.10	[81]
*Chlorella vulgaris*	**20**	38	88	84	36	84	48	21	55	1.25	[34]
*Chlorella vulgaris* *	18	37	93	48	**25**	95	45	23	60	1.10	[81]
*Chroomonas salina*	**18**	41	78	61	30	111	54	13	61	1.13	[48]
*Dunaliella bardawil*	18	42	110	70	35	95	54	7	58	1.06	[34]
*Dunaliella tertiolecta*	21	48	84	60	**18**	117	47	15	62	0.80	[48]
*Isochrisis aff.galbana (T-iso)*	**20**	46	87	60	31	105	45	16	61	1.25	[48]
*Isochrysis galbana*	21	48	87	62	**27**	108	52	13	62	1.15	[48]
*Nannochloris atomus*	18	34	75	52	**25**	94	40	11	59	1.07	[48]
*Nannochloropsis granulata (A)*	**23**	56	110	85	51	104	54	28	71	1.44	[54]
*Nannochloropsis oculata*	21	48	78	61	**20**	104	55	16	65	0.87	[48]
*Nitzschia closterium*	**14**	50	81	57	22	108	55	14	62	0.88	[48]
*Nostoc* sp.	20	**37**	95	65	38	140	53	10	72	1.23	[130]
*Pavlova lutheri*	50	49	81	**56**	37	111	43	15	67	1.17	[48]
*Pavlova salina*	**15**	44	90	62	20	92	52	9	61	0.94	[48]
*Phaeodactylum tricornutum*	**15**	46	70	64	42	82	48	26	51	0.94	[54]
*Phaeodactylum tricornutum*	**17**	49	77	56	23	107	54	16	59	1.06	[48]
*Pleurochrysis carterae*	**19**	42	99	72	44	154	57	11	76	1.18	[130]
*Porphyridium aerugineum*	**19**	71	119	80	59	121	58	33	73	1.19	[54]
*Scenedesmus obliquus*	**21**	36	73	56	21	80	51	**3**	60	0.45	[34]
*Skeletonema costatum*	**16**	52	83	57	26	109	51	13	63	1.00	[48]
*Spirulina platensis*	22	67	98	48	34	106	62	**3**	71	0.45	[34]
*Tetraselmis chuii*	18	35	75	57	**25**	91	42	10	58	1.07	[48]
*Tetraselmis chuii*	**16**	34	73	56	**52**	77	40	23	48	1.00	[54]
*Tetraselmis suecica*	**18**	35	80	60	30	97	41	12	57	1.13	[48]
*Thalassiosira pseudonana*	**16**	55	84	59	27	110	52	8.7	61	1.00	[48]

SAA—methionine and cystein; AAA—phenylananine and tyrosine. *—disrupted using microfluidizer. #—calculated by the authors of this review mathmatically, against the reference pattern of the essential amino acids for 3–10-year-old children (FAO/WHO/UNU, 2013). ^&^—the first limiting amino acid.

## Data Availability

Not applicable.

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
