# Peer review of "Microalgae as Sources of High-Quality Protein for Human Food and Protein Supplements"

_foods, 2021, doi:10.3390/foods10123002_

Round 1

Reviewer 1 Report

In general, the review is well written and it addresses an interesting and current topic. However, I would suggest to make the review more readers friendly by changing the structure, adding some figures and maybe some tables as well. The chapters were well planed however sometimes I find the information which are not relevant in that part, f.ex. In the chapter “Historical use of microalgae as human food: Authors write: “Proteins play an important role in the structure and metabolism of microalgal cells. They are an integral component of the membrane and light harvesting complex, including numerous catalytic enzymes involved in photosynthesis and carbon sequestration.”

Those are the only comments and suggestions I have.

Author Response

Thank you and please find below the attached our responses to your comments. 

Reviewer 2 Report

The review article on "Microalgae as sustainable sources of high quality protein for human food and protein supplements" is well written. 

The article becomes a monotonous read with no figures and table. Inclusion of atleast a table and figure would improve the article.

Author Response

Thank you and please find below our response to your comment. 

Comment: The article becomes a monotonous read with no figures and table. Inclusion of at least a table and figure would improve the article.

Response: A figure for the factors influencing protein content and quality of microalgae has been added on page 15. A table of protein content has been added on page 5, and another on the amino acid profile of some microalgae has been added on page 12.

Reviewer 3 Report

I believe the review in the present form does not add much to the existing synthesis on the topic, a greater literature mining work is needed to cover more features and more modern aspects (SUSTAINABILITY) of the topic as it is stated in the title...

to this purpose I recommend to report at least works from A. Vonshak as for biomass/protein production, Mario Tredici, Maria Hayes and Luisa Gouveia labs for microalgae protein in food application for nutraceutical and pharma applications...

I have included many other suggestions in the uploaded pdf to improve the manuscript readibility and organization 

Author Response

We appreciate your constructive comments and suggested changes. Please see below for our responses to your comments. 

1. I believe the review in the present form does not add much to the existing synthesis on the topic, a greater literature mining work is needed to cover more features and more modern aspects (SUSTAINABILITY) of the topic as it is stated in the title... to this purpose I recommend to report at least works from A. Vonshak as for biomass/protein production, Mario Tredici, and Luisa Gouveia labs for microalgae protein in food application for nutraceutical and pharma applications...

The title has been changed to “Microalgae as sources of high quality protein for human food and protein supplements” and deleted “sustainable”.  

The suggestions have been taken and accordingly added their work and references on pages 2-4.

2. I have included many other suggestions in the uploaded pdf to improve the manuscript readability and organization 

The authors revised the manuscript by addressing each comment and highlight. We have also replied to each comment in the attached PDF file provided by the reviewer. See the attached PDF file.

Reviewer 4 Report

The manuscript entitled "Microalgae as sustainable sources of high quality protein for human food and protein supplement" signed by Yanwen Wang and colleagues is a interesting review on the microalgae protein source for human consumption.

The general structure of the work is simply and well described, and the text is clear and concise.

In my opinion, some revisions are needed to make:

- Page 2, Line 88: please insert also carotenoids in the list of nutrient present in microalgae.

- Page 4, Line from 150-169: In my opinion could be more representative if the ranges of protein are insert in a table, reporting for each specie the protein content recorded in the previous studies.

- Page 7-8, section 7: In my opinion could be more clear if the formula to calculate ASS and PDCAAS are describes.  

Author Response

Thank you for taking time to review and comment on our manuscript for improvement. Please see below our responses.

1. Page 2, Line 88: please insert also carotenoids in the list of nutrient present in microalgae.

This has been taken and added.

 2. Page 4, Line from 150-169: In my opinion could be more representative if the ranges of protein are insert in a table, reporting for each specie the protein content recorded in the previous studies.

We modified this section by removing the lengthy list of specific algae species names found in refs. [37] and [51] and just giving the range of protein content as a % of biomass dry weight reported in those studies.

3. Page 7-8, section 7: In my opinion could be clearer if the formula to calculate ASS and PDCAAS are describes.  

The equations for AAS and PDCAAS have now been included.

Round 2

Reviewer 2 Report

All corrections included. The article can be accepted in the present form.

Reviewer 3 Report

Accepted in the present form